# Attitude of youth towards self-employment: Evidence from university students in Yemen

**Nabil Al-Qadasi** [1,2]*, **Gongyi Zhang**[1], **Ibrahim Al-Jubari** [3]

**1** Department of Business Administration, School of Management, Jilin University, Changchun, Jilin, China, **2** Faculty of Commerce and Economics, Sana'a University, Sana'a, Yemen, **3** Faculty of Business Management and Professional Studies, Management and Science University, Shah Alam, Malaysia

* alnabil1976@gmail.com

## Abstract

This study assesses whether final-year undergraduate students at Sana'a University, Yemen intend to start their own business. The study employs the theory of planned behaviour and two environmental factors to explore whether the theory's behavioural factors and the contextual factors of Lüthje & Franke's model have an impact on students' intentions to start their own business. A questionnaire survey with a random sample of 335 final-year university students from the largest public university in Yemen has been conducted. Data has been analysed using descriptive statistics, Pearson's correlation and structural equation modelling. The findings indicate that students' perceptions of entrepreneurship have a strong, direct impact on self-employment intention, excluding social norms and entrepreneurial self-efficacy. Students' self-employment intention is directly affected by perceived barriers and support factors in the entrepreneurship-related context. To increase their entrepreneurial abilities, university students require more training and education to be able to start new businesses. Developing entrepreneurial skills among citizens may improve the societal norms of business. The outcomes provide significant implications for policymakers, academic communities and international bodies.

## 1. Introduction

The role of entrepreneurship in economic development has received increasing attention since evidence has proven that entrepreneurship generates the most economic activity [1–3]. Countries with high unemployment rates where economic systems search for alternative methods to provide employment usually have economic institutions with focused policies to facilitate self-employment in the general population, especially for university graduates [4–6]. Today, the unemployment rate among the highly-educated population is proportionally more than the less-educated population [4, 7]. Therefore, providing job opportunities for all graduate students is a crucial issue of relevant authorities in various contexts. Directing the tendency of the youth to engage in entrepreneurship as a career option is considered to be an important strategy for reducing unemployment of university graduates in different countries. Several studies have indicated that entrepreneurship is a career interest in various countries [8–10], regions [11], several business administration programs and university students [12–14].

**Funding:** The publication fee of this research was funded by Jilin University. The funder had no role

in study design, data collection and analysis, decision to publish, or preparation of the manuscript.

**Competing interests:** The authors have declared that no competing interests exist.

To reduce the rate of unemployment among graduates, understanding their attitudes and perceptions regarding self-employment are key determine -economic behaviour and later design programs that can facilitate self-employment. Focus should be made on examining the most common contextual barriers as well as support factors that influence the youth's inclination to become self-employed [15–17]. It is important to direct the youth towards self-employment, encouraging them to become future entrepreneurs. The self-employment intention frequently begins prior to starting a business. Some studies on entrepreneurial intention have found a link between personal intentions and business start-up decisions [5, 18]. The self-employment decision depends on several motivational factors or attitudinal antecedents of intention. Motivational factors make self-employment more attractive, for instance, earning huge profits, gaining personal satisfaction as an entrepreneur and working at the location of choice. Such factors may attract and encourage young adults to start their own business [13, 19–22]. Inversely, contextual barriers are also present which make self-employment less attractive such as lack of capital, lack of skills, hard reality, etc. [13, 23–25]. Analysing the factors that influence the self-employment intention of the youth will assist policymakers in recognising and setting the relevant policies for fostering entrepreneurship and successfully integrating them into the labour market. However, the factors that influence self-employment intentions vary between countries, societies, cultures and individual perceptions [9, 26]. Empirical data is critically needed to determine the antecedents of entrepreneurial intention specific to each country. Globally, countries require youth who are innovative, modern, smart, determined, daring, efficient and employable. In other words, they must be 'entrepreneurial' to establish and run businesses.

Various contributions have used Ajzen's Theory of Planned Behaviour (TPB) [15] to explain individuals' self-employment intentions with three motivational factors: attitude towards the behaviour, subjective norms and perceived behavioural control [9, 19, 20, 27, 28]. Other studies have highlighted the contextual factors that may substantially encourage or discourage students from becoming self-employed, such as the perceived barriers and perceived support for entrepreneurship, based on Lüthje & Franke's Model (LFM) [13, 14]. However, not much evidence is present in the entrepreneurship literature on the motivational factors and the environmental relationship for starting a business [14, 25].

To fill the research gap and determine the youth's motivation towards self-employment intention as a career alternative, this study examines the motivational factors (attitude towards entrepreneurship, social norms and entrepreneurial self-efficacy) based on TPB as well as two environmental-related factors (perceived barriers and perceived support for entrepreneurship) based on LFM. The personality traits in Lüthje & Franke's model were not included; instead, Ajzen's [15] suggestion was followed to examine the effect of behaviour antecedents on the intention to start a business. This paper specifically aims to answer the following two questions:

Q1: How does attitude towards entrepreneurship, social norms and perceived entrepreneurial self-efficacy affects students' self-employment intention?

Q2: How do perceived barriers and perceived support of entrepreneurship affect students' self-employment intention?

## 2. Context of the study

Yemen is a Middle Eastern country located in the southern part of the Arab Peninsula. The estimated population is more than 29 million with a growth rate of 3%, one of the highest in the world. By 2030, Yemen's population is estimated to reach 38 million. The issue of

population growth is one of the most significant challenges facing the country's authority today. Yemen enjoys various natural resources such as oil & gas, tourism, historical attractions, long and beautiful beaches, fishing, diverse crops and human capital [29]. The government depends on the oil and gas sector as the major source of economic growth. However, six years of civil and regional war in Yemen has reversed decades of developmental progress. Violent conflict created grievous costs on the nation, damaging lives and properties. This has devastated the country's essential infrastructure, which was not in good shape even before the conflict. A majority of production activities had ceased since March 2015. Yemen is now considered to be the largest humanitarian crisis in the world. Although the official statistics on Yemen's economy are still scarce and unavailable, recent figures published by the World Bank suggest that the Yemen's GDP had contracted by 39% between 2014 and 2019 [30]. This reflects on how the country's economy is suffering. The statistics from the World Bank further revealed that the headcount poverty rate, according to the World Bank's International Poverty Line with daily per capita consumption of $USD 1.90 PPP, is projected to have increased from 33% during 2014 to approximately 52% in 2019. The unemployment rates were already high before the conflict. According to The Second National Millennium Development Goals Report–Yemen (MDGR, 2010) [31] conducted in 2010, the unemployment rate among the Yemeni youth has been estimated to be around 53%, which has worsened since then. Entrepreneurial activities are regarded as the driving force for creating strong job opportunities, making positive social change, providing support to the economy and speed up recovery from conflict. However, starting a business is very challenging in such times since the economic environment is uncertain. In 2009, a study by the Global Entrepreneurship Monitor (GEM) for Middle East and North Africa (MENA) indicated that Yemen ranked first among 55 economies regarding the proportion of the adult population who considers entrepreneurship as a desirable career choice. They believe entrepreneurs have high worth and respect in society [32]. The same study also reported that about 43% of Yemeni adults expressed the fear of failure when starting a business, while over 60% perceived they have the knowledge, skills and experience to start a business. Approximately 27% of Yemenis expect to start their own business in the next three years. Although the statistics indicate that entrepreneurship is a good career choice, it is unknown whether this still holds true in the current challenging conditions of the country.

## 3. Theoretical foundation and hypotheses development

### 3.1 Self-employment intention

In this study, self-employment intention is defined as an individual's willingness to start a new venture after graduation. Numerous studies have used self-employment as a tool for measuring the intention level of entrepreneurship [33]. Two intention-based models, Ajzen's Theory of Planned Behaviour (TPB) [15] and Shapero's Entrepreneurial Event Model (EEM) [34], are the most influential in terms of their ability to predict entrepreneurial intention. Krueger et al. would be the earliest scholars who compared and integrated existing theories of entrepreneurial intentions (TPB and EEM). They discovered that theories somewhat overlap [19]. Recently, Schlaegel & Koenig have tested and compared TPB and EEM to explain entrepreneurial intention. They confirmed that they are an integrated model of entrepreneurial actions. To date, these two models dominate entrepreneurial intention research [35].

According to EEM, entrepreneurial intentions depend on three elements: perceived desirability, perceived feasibility and propensity to act. Perceived desirability refers to the attractiveness of starting a business. Perceived feasibility refers to an individual's ability to start a business, while propensity to act refers to a person's disposition to act upon opportunities.

There are three independent motivational factors in TPB that influence intention. The first is personal attitude towards becoming self-employed, which refers to the expected value or utility of starting a business venture [21]. The second predictor of intention is social norms, which refers to the perceived social and cultural pressures felt by an individual regarding whether or not to perform entrepreneurial behaviours [36]. The third antecedent of intention is perceived behavioural control or perceived self-efficacy [37], which refers to the perceived ease or difficulty of becoming self-employed [38]. According to [15], TPB not only explains human behaviour, but also predicts it. A person's entrepreneurial behaviour refers to the result of the intention, and intention is a joint function of attitude, social norms and behavioural control. Self-employment is also a planned behaviour that cannot be created without adequate preparation. On the other hand, EEM focuses on the characteristics and previous entrepreneurial experiences of the individual. That is why we chose TPB to analyse the attitude of Yemeni youth towards self-employment intentions.

However, the environmental factors in both EEM and TPB do not directly affect intention or behaviour, therefore, we added two contextual factors which have a direct exogenous influence. These factors specifically focus on entrepreneurial intentions based on LFM, they are perceived barriers and perceived support. According to [39], LFM provides a well-articulated framework where we can assess the antecedents of self-employment intentions.

## 3.2 Attitude towards self-employment

Attitude towards self-employment refers to the expected efficiency of creating a new business, or the outcomes of starting a business. The outcomes of attitude/desirability towards self-employment represent a person's favourable or unfavourable valuation of a particular behaviour. Individuals develop attitudes from the beliefs they hold regarding the outcome of performing the behaviour [15].

These beliefs are positively related to attitude. Individuals generally prefer self-employment for various reasons, e.g., to make money [22], have autonomy [40], for the challenge [20], change [16], wealth [9], enjoying a certain quality of life [21], desire for increased independence [41], etc. These outcomes have been widely stated in the literature as the most important motivational factors for starting a business [18, 24]. They refer to the evaluation of self-employment that would prompt more efforts for individuals to engage in entrepreneurial activities [21, 42]. Value-based explanatory variables of self-employment intention should be carefully selected since beliefs are expected to vary between individuals. However, according to [43], students sometimes favour engaging in entrepreneurship for the usual reasons. In this study, four value-based variables, or four antecedents of attitude towards self-employment, are employed: attitude towards earning money, attitude towards personal satisfaction as an entrepreneur, attitude towards enhancing quality of life and lastly, attitude towards independence. Based on these four beliefs/outcomes for setting a business, the following hypothesis has been formulated:

**Hypothesis 1 (H1).** Attitudes toward entrepreneurship (earning money, gaining personal satisfaction as an entrepreneur, improving quality of life and establishing independence) have a positive influence on students' intention to become self-employed.

## 3.3 Perceived social norms

The second construct of TPB captures the social norms; whether or not to conduct a particular behaviour or, in our case, an entrepreneurial action. This is a critical determinant for the youth with no prior self-employment experience or limited entrepreneurial experience [21]. It further indicates that the perception of significant people (i.e., family, close friends, role

models, mentors or others) may further support the self-employement decision. Although social attitudes and perceptions play an important role in shaping individuals' entrepreneurial spirit, only a few studies have addressed the social valuation of entrepreneurship [1, 36, 44]. Accordingly, the following hypothesis has been suggested:

**Hypothesis 2 (H2).** Students surrounded by positive social norms regarding entrepreneurship will more likely intend to be self-employed.

## 3.4 Perceived behavioural control

Based on TPB, the third antecedent of intention is perceived behavioural control (PBC), which refers to the perceived ease or difficulty of performing a given behaviour [15]. In the field of entrepreneurship, PBC is termed entrepreneurial self-efficacy (ESE) as it is related to entrepreneurial intention. A high level of ESE should strengthen an individual's intention to perform the target behaviour [1]. ESE is defined as the degree to which individuals believe they have skills, knowledge and ability to start a new business [45]. Accordingly, the ability to create a business, operate a business, recognise opportunity, develop marketing & management skills and innovate may all contribute to Yemeni students' perceived behavioural control towards self-employment intention. Thus, the following hypothesis has been developed:

**Hypothesis 3 (H3).** Students who rate themselves higher in terms of entrepreneurial self-efficacy are more likely to have self-employment intentions.

## 3.5 Perceived barriers and support for entrepreneurship

Previous research that recognised the importance of external environmental factors on an individual's intention to engage in entrepreneurial activities provide some answers to the research questions. Several studies have primarily focused on the cultural barriers of entrepreneurship [10], the educational, structural and relational support affecting entrepreneurship [46, 47], the environmental-related barriers and support of entrepreneurship [13, 17], the image of entrepreneurs in society [48], the motivations and barriers hindering entrepreneurship [23, 41], the differences in social-cultural aspects [26, 49, 50], the entrepreneurial mentorship [51], a person's social networks [38], the socio-cultural norms [52], etc.

Empirical studies that linked the contextual factors of entrepreneurship with an individual's intention to be self-employed have provided mixed and inconsistent results. For instance, a study had found that curricular, regulatory environment and social environment resources all support entrepreneurial intentions [53]. Lu et al.'s study revealed that university support affects the intention of college students in creating a new venture [54]. Audretsch et al. claimed that the arrangement of institutions improve the entrepreneurship ecosystem in cities [55]. Othor studies have found that educational and structural support influence an individual's entrepreneurial intention [46, 56]. Lee et al. pointed out that each country must provide a customized entrepreneurship education that considers different cultural contexts [50].

On the other hand Elali & Al-Yacoub confirmed that social networks play a significant role in entrepreneurial intentions [38]. In contrast, Wang et al. reported that social networks have an indirect effect on the motivation for self-employment [57]. Begley et al. maintained that the social situation of entrepreneurship emerges as a good predictor of entrepreneurial interest [52].

Regarding the barriers, Choo & Wong reported that lack of capital, skills, confidence, hard reality, as well as compliant costs are considered to be the largest barriers of starting a business [23]. Giacomin et al. argued that the barriers differ according to country and culture [41] Lüthje & Franke emphasied that students' entrepreneurial intention is directly affected by perceived entrepreneurship-related barriers and support factors [17].

The partial inconsistency of results in previous studies indicate that it is still necessary to improve our knowledge of the contextual variables that may influence entrepreneurial intentions. To address the above gap, the current study focuses on the contextual factors that were assumed to have an influence on self-employment intention through two types of environmental-related factors: the perceived barriers and perceived support of entrepreneurship. The study hypothesises that individuals with unfavourable perceptions toward environmental-related factors are more likely to have weaker intentions to start a business. For instance, lack of capital, knowledge, support structure, confidence and hard reality all act as hindrances to an individual's intention of starting a business. In contrast, individuals who possess a positive attitude towards environmental-related support factors are more likely to have strong intentions of starting a business; for example, the positive influence of the university environment on student's entrepreneurial intentions, the support provided by the government and the backing of various economic institutions. Therefore, the following two hypotheses have been developed concerning entrepreneurship-related environmental factors:

**Hypothesis 4 (H4).** Students who perceive entrepreneurship-related barriers as more insurmountable are likely to have a weaker intention of starting a business.

**Hypothesis 5 (H5).** Students who perceive entrepreneurship-related support as more favourable are likely to have a stronger intention of starting a business.

Fig 1 presents the research framework. The motivational factors of TPB (attitude, social norms and perceived control) and environmental factors of LFM (perceived barriers and perceived support) predict the intention of final-year undergraduate students to become self-employed after graduation.

## 4. Methodology

### 4.1 Participants and sample size

The participants in this study are final-year university students of Sana'a University, a higher education institution in Yemen. It is the oldest and largest public university in the country, with approximately 79,460 of Yemen's 142,756 student enrolment. The university accounts for 36% of Yemen's student population. The university has 16 humanities and science faculties, 22 research centres as well as a wide variety of programs aimed at knowledge creation, dissemination and commercialisation. Eight faculties (0.50) were randomly selected to be considered as the research population, including commerce & economics, science, art, law, languages,

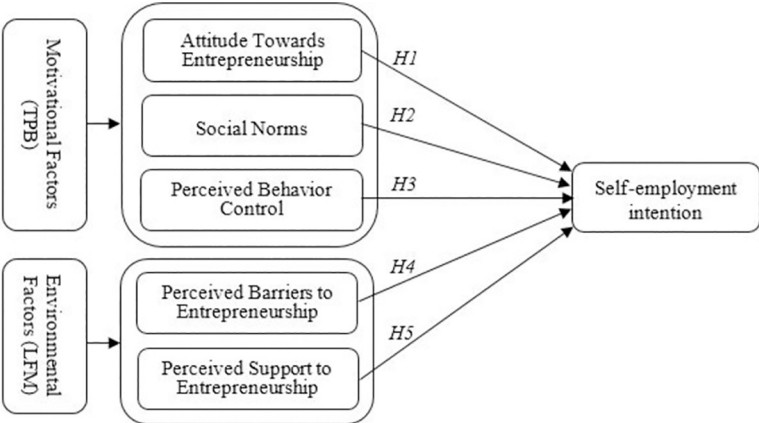

**Fig 1. The conceptual framework of self-employment intention.**

computer & IT, engineering and agriculture. This study assumes that Yemeni students perceive entrepreneurship as a desirable and feasible career option.

This study employed the stratified sampling technique. The required sample size was based on Thompson's the formula [58]: n = (N×p(1-p))/((N-1)(d^(2) ÷ z^2) +p(1-p)) where n = the required sample size, N = population size, z = confidence level at 0.95 (1.96), d = error proportion (0.05) and p = probability (0.50). Class representatives assisted in randomly distributing the questionnaires among the target respondents. Of the 370 questionnaires supplied, 335 were fully completed and returned, providing a response rate of 90.54%. The age of respondents ranged between 22 to 27 years old. The mean age is 23.24 years old (SD = 1.036). Approximately 64.18% are male and 35.82% are female. Table 1 provides further details regarding the sample distribution.

## 4.2 Questionnaire design

The Self-Employment Intention Questionnaire (SEIQ) was applied to assess the self-employment intentions of university students in Yemen based on the questionnaire used in previous studies [24, 26]. The questionnaire consists of 64 statements. All questionnaire statements relating to motivational and environmental features that influence self-employment intention were formulated as sentences. Questuons were answered on a 5-point Likert-scale ranging from 1 = 'strongly disagree' to 5 = 'strongly agree'. S1 Appendix presents the Self-Employment Intention Questionnaire (SEIQ) code, constructs, and items. All measures were developed in English and later translated to Arabic by one of the authors since that is the local language of respondents. The translated version was later reviewed by bilingual experts. Data was gathered form October to November during the 2020/2021 academic year using a self-administered class survey. Descriptive statistics, Pearson's correlation and structural equation modelling (SEM) were used for the data analysis.

## 4.3 Operationalisation of variables and measures

**Self-employment intention.** In this study, self-employment intention is the dependent variable. Four items developed by Liñán & Chen have been used to assess the students' self-

**Table 1. Sample distribution by gender, age, prior entrepreneurial experience and faculty.**

| Characteristics | Category | Frequency (N) | Frequency (%) |
|---|---|---|---|
| Gender | Male | 215 | 64.18 |
| | Female | 120 | 35.82 |
| Age | 22–23 | 211 | 63 |
| | 24–25 | 114 | 34 |
| | ≥ 26 | 10 | 3 |
| Entrepreneurial experience | Yes | 38 | 11.34 |
| | No | 297 | 88.66 |
| Faculty | Commerce & Eco. | 162 | 48.36 |
| | Law | 56 | 16.72 |
| | Art | 49 | 14.63 |
| | Engineering | 21 | 6.27 |
| | Computer & IT | 14 | 4.18 |
| | Languages | 13 | 3.88 |
| | Sciences | 12 | 3.58 |
| | Agriculture | 8 | 2.39 |

employment intention [26]. The sample item is "I am willing to do anything to become an entrepreneur/self-employed".

**Attitude towards entrepreneurship.** Attitude towards the expected value of starting a business was examined by collecting relevant data such as students' self-employment views and perception towards starting a business. The attitude was measured using four value-based explanatory variables taken from previous works [13, 59, 60]. Each variable includes three to four items. These explanatory variables influence the attitude towards entrepreneurship. They are: earn money, personal satisfaction, improve quality of life and the desire for independencel.

The 'earn money' variable is derived from scores of questions regarding the motivational importance of money as a criterion to assess personal success and solve financial problems. The sample item is "To me, high income is an important criterion in assessing the level of personal success".

The 'personal satisfaction' variable is based on the scores of questions regarding personal satisfaction as an entrepreneur and society's respect for the self-employed individual. The sample item is "Being an entrepreneur/self-employed would give me great satisfaction".

The 'improve quality of life' variable is derived from the belief that self-employment can increase personal and family quality of life. The sample item is "Being an entrepreneur/self-employed implies improving my quality of life".

The 'desire for independence' variable is derived from opinions regarding the importance of independence at work and respondents'preferences to work at the location of choice. The sample item is "Independence at work is important to me".

**Perceived social norms.** Like all previous studies, this construct was used to measure the perceived subjective norms and social pressures of starting a business. Three items have been taken from [26] to assess this variable. The sample item is "My family would approve of my decision to be self-employed". Two items have also been taken from [1] to measure social valuation. The sample item is "In my society, entrepreneurial activity is worthwhile despite the risks".

**Perceived behavioural control (PBC).** As previously mentioned, PBC is also known as entrepreneurial self-efficacy (ESE). This indicator was used to assess whether students possess the knowledge, skills and experience required to start a new business. Based on the previous work of Liñán & Chen [26], seven items have been applied to measure this construct. The sample item is "Opening and operating a business are easy/not difficult for me".

**Perceived barriers of entrepreneurship.** This construct was devoted to measure the perceived barriers of starting a business. Choo & Wong [23] identified five factors as barriers of establishing a business. The current study borrows these factors to measure the perceived barriers, they are: lack of capital, lack of knowledge, lack of confidence, lack of support structure and hard reality.

The 'lack of capital' factor was derived from scores obtained from the answered questions on the difficulties of finding capital providers, lack of personal savings and high-interest rates when applying for loans. Five items were used to measure this factor. The sample item is "Difficulties in obtaining finance".

The 'lack of knowledge, skills and experience' factor is derived from the answered questions on the issues of inadequate managerial, marketing, accounting, training and business experience. Five items were used to measure this factor. The sample item is "Lack of marketing knowledge and skills".

The 'hard reality' factor is derived from the answered questions concerning current economic-political conditions. Four items were used to measure this factor. The sample item is "Current economic/political conditions".

The 'lack of self-confidence' factor is derived from answered questions regarding fear of failure, lack of ideas on what type of business to start and difficulty in convincing others about an idea. Three items were used to measure this factor. The sample item is "Fear of failure".

The 'lack of support structure' factor is derived from answered questions regarding the difficulties of complying with government laws and regulations, high taxes and fees, lack of formal and informal help and bureaucratic procedures. Five items were used to measure this factor. The sample item is "Compliance with government regulations".

**Perceived support to entrepreneurship.**   This construct has been included in this study to capture the perceived support for entrepreneurship in Yemen. Three variables were used, with each variable possessing three to four items adapted from previous studies [46, 57, 61] such as: educational support, governmental support and institutional support.

The 'educational support' variable is derived from answered questions regarding the importance of the university's role in increasing entrepreneurial skills. Four items were used to measure this factor. The sample item is "The education in university improved my entrepreneurial spirit".

The 'governmental support' variable is derived from answered questions regarding the importance of governmental help and backing to start a new business. Four items were used to measure this factor. The sample item is "The government shows willingness to help individuals who want to be entrepreneurs/self-employed".

The institutional support variable is derived from answered questions regarding the importance of institutional encouragement and support for potential entrepreneurs in the context of this study. Three items were used to measure this factor. The sample item is "In Yemen, entrepreneurs are encouraged by an institutional structural system that includes private, public and non-governmental organisations".

## 5. Analysis and results

### 5.1 Confirmatory factor analysis (CFA)

To ensure the validity and reliability of the measurements, the researchers employed confirmatory factor analysis (CFA) using SPSS, Amos software version 22. According to the guidelines of previous studies [62, 63], we applied the following criteria for the Goodness-of-Fit: i) the standardised regression weight values (factor loading) should be above 0.50, ii) the average of variance extracted (AVE) should be above 0.50 and iii) the composite reliability and Cronbach's α coefficients should be above 0.70.

The CFA results using the maximum likelihood estimation on the covariance matrix revealed that the standardised regression weights (factor loading) of items used in the current study ranged between 0.571 and 0.912. The AVE of factors was above 0.50 and ranged from 0.549 and 0.757. The composite reliability values for all included measures used in the study presented a high degree of consistency and reliability ranging from 0.738 to 0.909. Lastly, the Cronbach's α coefficients and reliability values for all factors were acceptable and above 0.732 in each case, indicating that construct reliability and validity have been achieved. Table 2 lists the results of the measurement model.

### 5.2 Structural equation modelling (SEM)

Structural Equation Modelling (SEM) has been widely used in social sciences research in previous years [64]. The covariance-based structural equation modeling (CB-SEM) with maximum like-lihood estimation was employed to analyse the data. The CB-SEM is deemed more appropriate for the assessment of the study model's adequacy and quality, and can further test hypotheses using the AMOS program.

**Table 2. Measurement model.**

| First-order factors | Second-order factor | Items | SFL | AVE | CR | Cronbach's α |
|---|---|---|---|---|---|---|
| Attitude towards entrepreneurship | Earn money | ATEM1 | 0.907 | 0.673 | 0.802 | 0.791 |
| | | ATEM2 | 0.724 | | | |
| | Personal satisfaction | ATPS1 | 0.755 | 0.653 | 0.848 | 0.845 |
| | | ATPS2 | 0.903 | | | |
| | | ATPS3 | 0.758 | | | |
| | Improving quality of life | ATQL1 | 0.756 | 0.716 | 0.909 | 0.887 |
| | | ATQL2 | 0.908 | | | |
| | | ATQL3 | 0.856 | | | |
| | | ATQL4 | 0.859 | | | |
| | Desire for independence | ATDI1 | 0.841 | 0.702 | 0.875 | 0869 |
| | | ATDI2 | 0.911 | | | |
| | | ATDI4 | 0.756 | | | |
| Self-employment intention | | SEI1 | 0.835 | 0.619 | 0.866 | 0.867 |
| | | SEI2 | 0.823 | | | |
| | | SEI3 | 0.758 | | | |
| | | SEI4 | 0.728 | | | |
| Social norms | | SN1 | 0.865 | 0.549 | 0.782 | 0.822 |
| | | SN2 | 0.650 | | | |
| | | SN4 | 0.692 | | | |
| Perceived behavioral control | | PBC1 | 0.828 | 0.669 | 0.890 | 0.882 |
| | | PBC2 | 0.848 | | | |
| | | PBC3 | 0.803 | | | |
| | | PBC4 | 0.794 | | | |
| Perceived barriers to entrepreneurship | Lack of capital | PBLC1 | 0.906 | 0.619 | 0.825 | 0.802 |
| | | PBLC2 | 0.844 | | | |
| | | PBLC3 | 0.571 | | | |
| | Lack of knowledge | PBLK1 | 0.823 | 0.723 | 0.839 | 0.839 |
| | | PBLK2 | 0.877 | | | |
| | Hard reality | PBHR2 | 0.863 | 0.757 | 0.903 | 0.902 |
| | | PBHR3 | 0.912 | | | |
| | | PBHR4 | 0.835 | | | |
| | Lack of cnfidence | PBSC1 | 0.874 | 0.721 | 0.885 | 0.882 |
| | | PBSC2 | 0.809 | | | |
| | | PBSC3 | 0.863 | | | |
| | Lack of support structure | PBLS2 | 0.839 | 0.656 | 0.851 | 0.849 |
| | | PBLS3 | 0.830 | | | |
| | | PBLS5 | 0.760 | | | |
| Perceived support to entrepreneurship | Educational support | PES1 | 0.862 | 0.751 | 0.900 | 0.901 |
| | | PES2 | 0.872 | | | |
| | | PES3 | 0.867 | | | |
| | Institutional support | PIS1 | 0.681 | 0.635 | 0.837 | 0.831 |
| | | PIS2 | 0.797 | | | |
| | | PIS3 | 0.899 | | | |
| | Governmental support | PGS3 | 0.688 | 0.587 | 0.738 | 0.732 |
| | | PGS4 | 0.840 | | | |

Notes: **SFL–standardised factor loading; AVE–average variance extracted; CR–composite reliability;** items (ATEM 3, ATEM 4, ATPS 4, ATDI 3, SN 3, SN 5, PBC 5, PBC 6, PBC 7, PBLC 4, PBLC 5, PBLK 3, PBLK 4, PBLK 5, PBHR 1, PBLS 1, PBLS 4, PES 4, PGS 1, PGS 2) were deleted for the goodness of fit of the model.

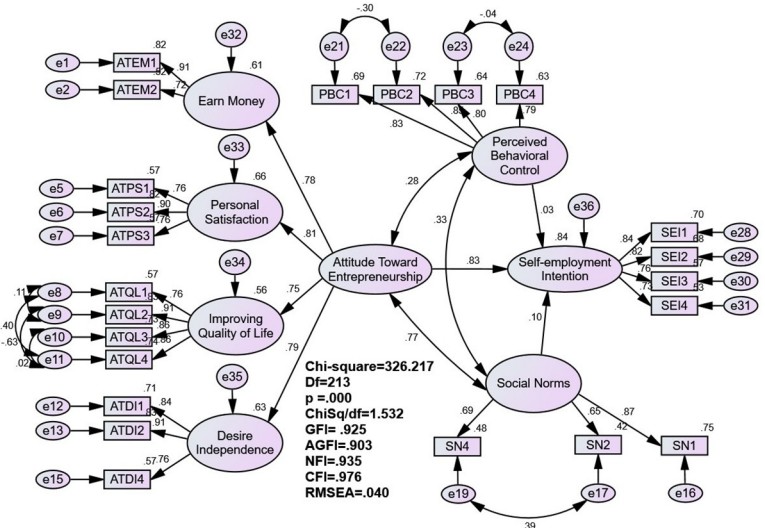

**Fig 2. SEM analysis of the motivational factors of self-employment intention.**

Two SEM models have been built in this study, presented in Figs 1 and 2. Fig 1 is the base model according to Ajzen's (TPB) [15], while Fig 2 is based on Lüthje & Franke's Model (LFM) [17]. Previous studies were used as guides for distinguishing these models [21, 65]. The base model predicts that attitudes concerning entrepreneurship, social norms and perceived behavioural control are positively associated with students' self-employment intention. The second model predicts that perceived barriers and perceived support for entrepreneurship are associated with students' self-employment intention. The TPB's hypothesised measurement model includes 32 indicators that reveal seven unobserved latent constructs: attitude towards earning money, attitude towards personal satisfaction, attitude towards improving quality of life, attitude towards the desire for independence, social norms, perceived behavioural control and self-employment intention.

The initial measurement model yielded an unsatisfactory goodness-of-fit values. Two items from attitude towards earning money, one from attitude towards personal satisfaction, one from attitude towards the desire for independence, three from perceived behavioural control and two from social norms have all contributed to a poor fitting model, thus, they were removed from the analysis (i.e., ATEM 3, ATEM 4, ATPS 4, ATDI 3, PBC 5, PBC 6, PBC 7, SN 3 and SN 5). The revised measurement model, displayed in Table 3 presents a better alternative

**Table 3. Goodness-of-fit indices of the model.**

| Fit indices | Model 1 (TPB model) | Model 2 (LFM model) | Suggested values | Supporting literature |
|---|---|---|---|---|
| $\chi^2$ | 326.217 | 372.881 | | |
| $\chi^2$/df | 1.532 | 1.295 | >1 and <5 | [66] |
| GFI | 0.925 | 0.923 | $\geq$ 0.90 | [67] |
| AGFI | 0.903 | 0.907 | $\geq$ 0.90 | [68] |
| NFI | 0.935 | 0.919 | $\geq$ 0.90 | [69] |
| CFI | 0.976 | 0.980 | $\geq$ 0.90 | [70] |
| RMSEA | 0.040 | 0.030 | $\leq$ 0.05 | [71] |

Notes: χ2 –CMIN; df–degree of freedom; GFI–goodness-of-fit index; AGFI–adjusted goodness-of-fit index; NFI–normal fit index; CFI–comparative fit index; RMSEA–root mean square error of approximation.

and fulfils the acceptance criteria with the following indices: $\chi^2 = 326.217$, $\chi^2/df = 1.532$, GFI = 0.925, AGFI = 0.903, NFI = 0.935, CFI = 0.976 and RMSEA = 0.040.

Similarly, the hypothesised measurement model of LFM included 37 indicators with nine unobserved latent constructs: the perceived barriers of lack of capital, lack of knowledge, hard reality, lack of self-confidence, lack of support structure, educational support, government support, institutional support and self-employment intention.

The initial measurement model yielded an unsatisfactory goodness-of-fit values. Two items from lack of capital, three from lack of knowledge, one from hard reality, two from lack of support structure, one from educational support and two from governmental support have all contributed to a poor fitting model, thus, they were removed from the analysis (i.e., PBLC 4, PBLC 5, PBLK 3, PBLK 4, PBLK 5, PBHR 1, PBLS 1, PBLS 4, PES 4, PGS 1 and PGS 2). The revised measurement model, displayed in Table 3, presents a better model and fulfils the acceptance criteria with the following indices: $\chi^2 = 372.881$, $\chi^2/df = 1.295$, GFI = 0.923, AGFI = 0.907, NFI = 0.919, CFI = 0.980 and RMSEA = 0.030.

Table 4 presents the descriptive statistics and inter-correlations between the constructs included in the study. The mean values indicate that all constructs have moderate to relatively high values. The perceived support for entrepreneurship variables acquired the lowest mean values. Institutional support was the lowest value (2.0328). The correlation between the study variables and self-employment intention is significant. The relationship between self-employment intention and attitude towards entrepreneurship and social norms is especially high. The relationship between self-employment intention and other study variables ranged between low to medium [72].

**Table 4. Means scores, standard deviation and correlations between the study variables.**

| | | Mean | SD | 1 | 2 | 3 | 4 | 5 | 6 | 7 | 8 | 9 | 10 | 11 | 12 | 13 | 14 | 15 |
|---|---|---|---|---|---|---|---|---|---|---|---|---|---|---|---|---|---|---|
| 1 | ATEM | 3.9627 | 0.68841 | 1 | | | | | | | | | | | | | | |
| 2 | ATPS | 3.8299 | 0.76988 | .536** | 1 | | | | | | | | | | | | | |
| 3 | ATQL | 3.8351 | 0.74162 | .482** | .554** | 1 | | | | | | | | | | | | |
| 4 | ATDI | 3.8761 | 0.67982 | .589** | .540** | .567** | 1 | | | | | | | | | | | |
| 5 | SNSV | 3.9982 | 0.84364 | .431** | .491** | .487** | .538** | 1 | | | | | | | | | | |
| 6 | PBC | 3.6320 | 0.82517 | .190** | .200** | .196** | .219** | .274** | 1 | | | | | | | | | |
| 7 | PBLC | 3.9081 | 0.71387 | .108* | .206** | .160** | .173** | .128* | .082 | 1 | | | | | | | | |
| 8 | PBLK | 4.0263 | 0.70416 | -.153** | -.161** | -.200** | -.193** | -.156** | -.156** | -.058 | 1 | | | | | | | |
| 9 | PBHR | 3.8187 | 0.82244 | .173** | .225** | .244** | .261** | .225** | .042 | .124* | -.047 | 1 | | | | | | |
| 10 | PBSC | 3.6627 | 0.89724 | -.151** | -.165** | -.085 | -.061 | -.114* | -.158** | -.018 | .255** | .041 | 1 | | | | | |
| 11 | PBLS | 3.9278 | 0.71856 | .125* | .193** | .199** | .215** | .152** | .068 | .177** | -.059 | .232** | -.062 | 1 | | | | |
| 12 | PES | 2.4485 | 0.89918 | .216** | .205** | .304** | .280** | .207** | .221** | -.034 | -.210** | .176** | -.098 | .112* | 1 | | | |
| 13 | PGS | 2.0933 | 0.68511 | .078 | .033 | .066 | .062 | .094 | .005 | .039 | .101 | .006 | .019 | .017 | -.014 | 1 | | |
| 14 | PIS | 2.0328 | 0.80454 | .164** | .125* | .122* | .170** | .235** | .063 | .091 | -.024 | .037 | -.014 | -.057 | .062 | .260** | 1 | |
| 15 | SEI | 3.9448 | 0.91883 | .602** | .629** | .659** | .622** | .573** | .258** | .164** | -.211** | .272** | -.158** | .203** | .304** | .109* | .241** | 1 |

Notes: The variables are labeled as follows; ATEM—attitude towards earning money; ATPS—attitude towards personal satisfaction; ATQL—attitude towards improving quality of life; ATDI—attitude towards the desire for independence; SN—aocial norms; PBC—perceived behavioral control; PBLC—perceived barriers of lack of capital; PBLK—perceived barriers of lack of knowledge; PBHR—perceived barriers of hard reality; PBSC—perceived barriers of lack of self-confidence; PBLS—perceived barriers of lack of support structure; PES–perceived educational support; PGS–perceived governmental support; PIS–perceived institutional support; SEI—self-employment intentions.

*p < 0.05

**p < 0.01 (n = 335).

## 5.3 Hypothesis testing

The study employed SEM to test and confirm the hypotheses (H1, H2, H3, H4 and H5), of the structural models.

H1 states that attitudes towards entrepreneurship (earn money, personal satisfaction as an entrepreneur, improve quality of life and independence) have a positive influence on students' intention to become self-employed. The study outcomes presented a significant impact between attitude towards entrepreneurship and students' self-employment intentions ($\beta$ = 0.829, t = 8.609, p < 0.001). As a result, H1 is supported. The four beliefs for setting a business work together to shape students' attitude towards entrepreneurship: earn money ($\beta$ = 0.781), personal satisfaction ($\beta$ = 0.814), improve quality of life ($\beta$ = 0.749) and the need for economic independence ($\beta$ = 0.791).

H2 postulates that students surrounded by positive social norms concerning entrepreneurship will more likely intend to be self-employed. However, the findings did not prove this assumption. The effect of social norms on students' self-employment intentions was insignificant ($\beta$ = 0.103, t = 1.225, p > 0.05). Thus, H2 is not supported.

H3 states that students who rate themselves higher in terms of entrepreneurial self-efficacy are more likely to have self-employement intentions. The findings revealed that students' perceived behavioural control or entrepreneurial self-efficacy does not have a significant impact on their self-employment intentions ($\beta$ = 0.027, t = 0.658, p > 0.05). Thus, H3 is not supported. Further details of the model estimates are presented in Table 5 and Fig 2.

H4 emphasises that students who perceive entrepreneurship-related barriers as insurmountable are more likely to have a weaker intention of becoming self-employed. The findings of the study support this idea. Students who view the environment as difficult for business

**Table 5. Hypotheses estimation.**

| H. No | Paths | Standardised regression weight | Regression weight | S.E | t-value | Findings |
|---|---|---|---|---|---|---|
| H1 | ATE—> EM | 0.781 | 1 | | | S |
| | ATE—> PS | 0.814 | 0.826 | 0.075 | 10.976 (<0.001) | |
| | ATE—> QL | 0.749 | 0.973 | 0.089 | 10.930 (<0.001) | |
| | ATE—> DI | 0.791 | 0.744 | 0.068 | 10.872 (<0.001) | |
| | ATE—> SEI | 0.829 | 1.248 | 0.144 | 8.656 (<0.001) | |
| H2 | SN—> SEI | 0.103 | 0.150 | 0.119 | 1.264 (>0.05) | NS |
| H3 | PBC—> SEI | 0.027 | 0.028 | 0.042 | 0.658 (>0.05) | NS |
| H4 | PBE—> LK | -0.318 | -0.949 | 0.324 | -2.930 (<0.01) | S |
| | PBE—> LC | 0.297 | 0.856 | 0.280 | 3.055 (<0.01) | |
| | PBE—> HR | 0.429 | 1.603 | 0.370 | 3.698 (<0.001) | |
| | PBE—> SC | -0.210 | -0.672 | 0.279 | -2.408 (<0.05) | |
| | PBE—> LS | 0.387 | 1 | | | |
| | PBE—> SEI | 0.679 | 2.441 | 0.668 | 3.652 (<0.001) | |
| H5 | PSE—> ES | 0.145 | 0.298 | 0.180 | 1.655 (>0.05) | PR |
| | PSE—> GS | 0.470 | 0.844 | 0.296 | 2.848 (<0.01) | |
| | PSE—> IS | 0.695 | 1 | | | |
| | PSE—> SEI | 0.344 | 0.706 | 0.278 | 2.538 (<0.05) | |

Notes: ATE–attitude towards entrepreneurship; EM–earn money; PS—personal satisfaction; QL—quality of life; DI—desire for independence; SN—social norms; PBC—perceived behavioural control; PBE—perceived barriers of entrepreneurship; LC—lack of capital; LK—lack of knowledge, HR—hard reality; SC -lack of self-confidence; LS—lack of support structure; PSE–perceived support for entrepreneurship; ES—educational support; GS—governmental support; IS—institutional support; SEI—self-employment intention; S–supported; NS–not supported; PR–partially supported.

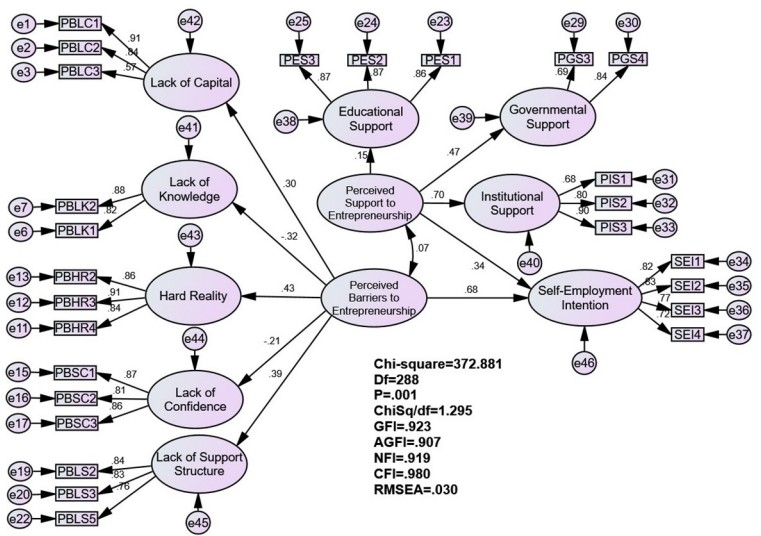

**Fig 3. SEM analysis of the environmental factors of self-employment intention.**

founders (such as financial problems, difficulties in accessing business information, hard reality and lack of support structure) are likely to have weaker intentions of starting a business ($\beta$ = 0.679, t = 3.652, p < 0.001). Based on this result, H4 is supported. Five environmental barriers of entrepreneurship work together to shape the perceptions of students: lack of capital ($\beta$ = 0.297), lack of knowledge ($\beta$ = -0.318), hard reality ($\beta$ = 0.429), lack of confidence ($\beta$ = -0.210) and lack of support structure ($\beta$ = 0.387).

H5 postulates that students who perceive entrepreneurship-related support as more favourable are likely to have a stronger intention of starting a business. A supportive environment for entrepreneurship (e.g., educational, governmental and institutional backing) could strengthen students' self-employment intentions. The study results completely prove this assumption ($\beta$ = 0.344, t = 2.538, p < 0.05). Hence, H5 is supported. Three types of entrepreneurship-related support work together to shape students' self-employment intentions: educational ($\beta$ = 0.145), governmental ($\beta$ = 0.470) and institutional ($\beta$ = 0.695). Further details of the model estimates are presented in Table 5 and Fig 3.

## 6. Discussion and conclusion

As previously mentioned, this study examined students' intentions of starting their own business, clarifying the contextual factors that may influence their intentions of becoming future entrepreneurs. In the current research, the TPB model explained 84% of the variance in self-employment intention. This paves the way for additional research concerning value-based variables of entrepreneurship (wealth, prestige, quality of life, satisfaction, autonomy, etc.).

Our findings indicate that the first construct of TPB, attitude towards entrepreneurship, has a significant impact on students' self-employment intentions. Attitude towards earning money, personal satisfaction as an entrepreneur, improving quality of life and the need for independence are the four most crucial factors for measuring attitude concerning entrepreneurship [16, 19–22]. As suggested by Krueger et al. [19], the expected outcomes of starting a business (such as autonomy, stress, financial performance, personal satisfaction and personal quality of life) make individuals more likely to start their own business. Many scholars confirm that attitude is the most significant predictor of entrepreneurial intentions [1, 16, 27, 73].

However, Zhang et al. [21] found the surprising result that attitude has no effect on entrepreneurial intention.

The second major construct of intention, based on the TPB model, is perceived social norms. Our findings surprisingly indicate that social norms have an insignificant impact on students' self-employment intention, which reflects on the perceived general acceptability of entrepreneurship as a favourable career path. In literature, several scholars had found a negative relationship between social norms and entrepreneurial intention [19, 22, 26, 74]. Ajzen [75] did suggest that perceived social pressure from immediate family, relatives, close friends and significant others may not be a powerful driver of intentions for individuals who have strong internal control. According to [1, 76], social norms and social support are antecedents to the attitude concerning behaviour and perceived behavioural control. However, in the current study, other social and cultural factors may influence students' self-employment intentions instead of the opinions of immediate family, close friends, teachers and significant others.

The third antecedent of intention based on the TPB model is students' entrepreneurial self-efficacy. Our results revealed that perceived entrepreneurial self-efficacy has an insignificant effect on students' self-employment intentions. Shah & Soomer found that in Pakistan, the perceived behavioural control factor is not positively related to entrepreneurial intentions [76]. Ferreira et al. studied the role of behavioural and psychological approaches that lead to entrepreneurial intention and found that the path between perceived behavioural control and entrepreneurial intention is insignificant [77]. However, in our study, perceived behavioural control was not a predictor of self-employment intention among final-year undergraduate students in Yemen. It may negatively reflect on the perceived capacity of Yemeni students to perform the target behaviour (in our case, starting a business) due to the uncertain business future brought on by difficult conditions as well as political and economic crises. On the other hand, it may also indicate that students have limited perception and understanding of the complexity or ease of engaging in entrepreneurial behaviour.

In this study, the importance of environmental factors in self-employment intention was also identified. Entrepreneurship-related barriers and support are significantly related to intentions. If students have a pessimistic outlook regarding the business environment (e.g., difficulty in obtaining finance, lack of business information, lack of formal or non-formal help to start a business and difficulty in complying with government regulations), they are less likely to become self-employed. In contrast, optimistic evaluation of formal or non-formal support, which are available to potential entrepreneurs, is linked to a stronger intention of starting a business and pursuing a career as an entrepreneur. In our investigation on self-employment intention of final-year undergraduate students, H4 and H5 state that the perceived barriers and support to entrepreneurship directly contribute to shaping students' self-employment intention. Overall, our results support both hypotheses. This is in line with the previous work of Lüthje & Franke [17] who confirmed that the factors have a direct impact on entrepreneurial intention. Kallas [25] found that a greater satisfaction with the external environment leads to stronger entrepreneurship intention. Choo & Wong [23] also confirmed that the barriers of starting a business for non-starters include hard reality, lack of capital, lack of skills, lack of confidence and compliancy costs.

Entrepreneurial intentions without action are worthless, particularly during high unemployment rates, as in the study's context. To exploit the positive attitudes toward entrepreneurship and translate intentions into action, potential entrepreneurs should have access to essential resources. Access to entrepreneurship support should also increase. Entrepreneurship education and training can help improve the perceived capabilities of starting a business. The appreciation and recognition of entrepreneurship as a viable career path in society and the

celebration of its successe can help enhance the societal norms toward entrepreneurship. In summary, today's youth are ready and willing to start new businesses, but they are unable to translate their entrepreneurial aspirations into more intentional planning and preparation.

## 7. Research limitations and future research directions

The present research is not without limitations, which should be addressed in future investigations. Although the findings are relevant to final-year undergraduate students, participants in this study were from business and non-business majors. Future research should solely focus on undergraduate and postgraduate students in business-related majors to investigate the effect of entrepreneurship education on participants' self-employment intentions. Secondly, the research is cross-sectional and not longitudinal. Changes in self-employment intention after graduation and engagement in self-employment activities should be measured. Thus, to generate additional insights regarding self-employment intentions, future studies should focus on measuring self-employment intentions and engagement in entrepreneurship activities before and after graduation. Thirdly, in war-torn societies, entrepreneurship may be influenced differently by various attributes and contexts. Thus, it is necessary to overcome the constraints of assessing this factor and to determine more accurate measurements. Finally, entrepreneurship is a very complex process affected by multiple variables and dynamics. The self-employment intention of university students is not easy to construct. The present research only examined a few of these factors by integrating motivational and contextual aspects to entrepreneurship in one model. Future research should also consider other motivational and contextual factors when investigating entrepreneurial intentions.

## Supporting information

**S1 Appendix. Self-employment intention questionnaire.** Code, constructs and items.
(DOCX)

**S1 Dataset. Data used in the study.**
(XLSX)

## Author Contributions

**Conceptualization:** Nabil Al-Qadasi.

**Data curation:** Nabil Al-Qadasi.

**Formal analysis:** Nabil Al-Qadasi, Ibrahim Al-Jubari.

**Investigation:** Nabil Al-Qadasi, Ibrahim Al-Jubari.

**Methodology:** Nabil Al-Qadasi.

**Project administration:** Nabil Al-Qadasi.

**Software:** Nabil Al-Qadasi.

**Supervision:** Gongyi Zhang.

**Writing – original draft:** Nabil Al-Qadasi.

**Writing – review & editing:** Gongyi Zhang, Ibrahim Al-Jubari.

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
