## [Decision Letter · Decision Letter 0]

12 Jul 2021

PONE-D-21-17082

Attitude of Youth Towards Self-Employment: Evidence from University Students in Yemen

PLOS ONE

Dear Dr. NABIL,

Thank you for submitting your manuscript to PLOS ONE. After careful consideration, we feel that it has merit but does not fully meet PLOS ONE’s publication criteria as it currently stands. Therefore, we invite you to submit a revised version of the manuscript that addresses the points raised during the review process.

The paper approaches an interesting theme, while the analysis is conducted in a comprehensive manner. Some minor changes are needed as required in the reviewers' comments listed at the bottom of this email.

We look forward to receiving your revised manuscript.

Kind regards,

Camelia Delcea

Academic Editor

PLOS ONE

Journal Requirements:

Reviewers' comments:

Reviewer's Responses to Questions

**Comments to the Author**

1. Is the manuscript technically sound, and do the data support the conclusions?

Reviewer #1: Yes

Reviewer #2: Yes

2. Has the statistical analysis been performed appropriately and rigorously? 

Reviewer #1: Yes

Reviewer #2: Yes

3. Have the authors made all data underlying the findings in their manuscript fully available?

Reviewer #1: No

Reviewer #2: Yes

4. Is the manuscript presented in an intelligible fashion and written in standard English?

Reviewer #1: Yes

Reviewer #2: No

5. Review Comments to the Author

Reviewer #1: The paper addresses an interesting topic. The article is easy to follow and the results are supported by the appropriate statistical methodology. However, I have some comments.

1) The sentence in lines 43-44 "Currently, the rate of unemployment among higher-educated people remains proportionally more than the rate of less-educated workers" must be motivated by data with sources or references;

2) Chapters must be enumerated to make the paper easier to follow;

3) For Chapter 2 (I guess) I would consider two separate bigger sub-section in this case: one for the behavioral factors and another one for barriers and environmental factors. In principle, the sub-section splitting should follow figure 1;

4) Why not merging the SEM subsection with the subsequent one? There are already too many sub-sections in the paper;

5) Overall, more technical details about the SEM can be provided;

6) The reference to figures 2 and 3 should be provided in line 393;

7) The dataset must be publicly available within either the authors website or a public data repository;

8) Perhaps 83 references are too much. I think that the authors can reduce the references only to the necessary ones.

Reviewer #2: Dear Author (s);

I can say that the paper is well structured and anyone is able to get an idea of the topic in question.

As minor point I think that some parts are not well structured at all levels of accuracy and an audience with a high level of understanding of the subject may not always find it pleasant.

I believe that once these problems have been fixed this paper can be a highly appreciated read by the readers of the journal and much requested in the context of attitude of youth toward self-employment.

So this is my first suggestion for you.

1) The "Methodology" paragraph should be longer and more accurate, highlighting better what has been achieved in the paper and what are the goals that could be achieved in the future. This in order to simplify the successful full understanding of the text by reminding the reader of the logical journey that has been taken while reading the paper.

2) There is no clear and ordered reference to the existing literature and methodologies currently in use or to the problems that prevent their full interpretation. I would have given a more schematic and complete visually ordered view to the literature to give a greater basis of credibility to the topic discussed.

3) Finally the English language should be checked by a native speaker.

Best Regards

6. PLOS authors have the option to publish the peer review history of their article (what does this mean?). If published, this will include your full peer review and any attached files.

Reviewer #1: No

Reviewer #2: No

---

## [Author Response · Author response to Decision Letter 0]

24 Jul 2021

Reviewer 1: we have incorporated all of your suggestions into our revision. They were valuable suggestion. Thank you.

Reviewer 2: we have incorporated all of your suggestions into our revision. They were very helpful. Thank you for your help.

---

## [Editor Report · Decision Letter 1]

31 Aug 2021

Attitude of Youth Towards Self-Employment: Evidence from University Students in Yemen

PONE-D-21-17082R1

Dear Dr. NABIL,

We’re pleased to inform you that your manuscript has been judged scientifically suitable for publication and will be formally accepted for publication once it meets all outstanding technical requirements.

Kind regards,

Camelia Delcea

Academic Editor

PLOS ONE
---

## [Editor Report · Acceptance letter]

3 Sep 2021

PONE-D-21-17082R1 

Attitude of youth towards self-employment: Evidence from university students in Yemen 

Dear Dr. Al-Qadasi:

I'm pleased to inform you that your manuscript has been deemed suitable for publication in PLOS ONE. Congratulations! Your manuscript is now with our production department. 

Kind regards, 

on behalf of

Dr. Camelia Delcea 

Academic Editor

PLOS ONE